

# N-acetylcysteine alleviates PCB52-induced hepatotoxicity by repressing oxidative stress and inflammatory responses

Wen-Tao Zhou[1], Li-Bin Wang[1], Hao Yu[2], Kai-Kai Zhang[3],
Li-Jian Chen[3], Qi Wang[3] and Xiao-Li Xie[1]

[1] Department of Toxicology, School of Public Health, Southern Medical University (Guangdong
Provincial Key Laboratory of Tropical Disease Research), Guangzhou, China
[2] The 2015 Class, 8-Year Program, The First Clinical Medical School, Southern Medical
University, Guangzhou, China
[3] Department of Forensic Pathology, School of Forensic Medicine, Southern Medical University,
Guangzhou, China

Corresponding authors
Qi Wang,
wangqi_legmed@vip.163.com
Xiao-Li Xie,
xiexiaoli1999@smu.edu.cn

## ABSTRACT

Polychlorinated biphenyls (PCBs), particularly low chlorinated congeners in our environment, can induce human hepatotoxicity. However, the mechanisms by which PCBs cause hepatotoxicity remain elusive. Moreover, there are no effective treatments for this condition. In this study, 40 µM PCB52 was administered to rat (Brl-3A) and human hepatocytes (L-02) for 48 h following the N-acetylcysteine (NAC)/saline pretreatment. A significant decrease in cell viability was observed in PCB52-treated cells relative to the control. Besides, PCB52 significantly increased reactive oxygen species (ROS) levels and malondialdehyde (MDA) contents, suggesting induction of oxidative stress. The expression of *Traf6*, *MyD88*, and *Tnf* in Brl-3A cells and that of *MYD88, TNF*, and *IL1B* in L-02 cells were significantly upregulated by PCB52. Consistently, overexpression of TLR4, MyD88, Traf6, and NF-κB p65 proteins was observed in PCB52-treated cells, indicating activation of inflammatory responses. Nevertheless, no changes in kelch-like ECH-associated protein 1 (keap1), nuclear factor-erythroid 2-related factor (nrf2), and heme oxygenase-1 proteins were observed in PCB52-treated cells, indicating non-activation of the keap1/nrf2 pathway. Pretreatment with NAC significantly ameliorated PCB52 effects on cell viability, ROS levels, MDA contents and expression of inflammatory elements at both RNA and protein levels. However, no changes in keap1, nrf2 and HO-1 protein levels were detected following NAC pretreatment. Taken together, with non-activated keap1/nrf2 pathway, PCB52-induced oxidative stress and inflammatory responses could be responsible for its hepatotoxicity. These effects were effectively attenuated by NAC pretreatment, which scavenges ROS and dampens inflammatory responses. This study might provide novel strategies for the treatment of the PCBs-associated hepatotoxic effects.

## INTRODUCTION

Polychlorinated biphenyls (PCBs) consist of a class of persistent chlorinated pollutants. Theoretically, PCBs are categorized into 209 congeners, including dioxin-like (DL) PCBs and non-DL (NDL) PCBs (*Zwierello et al., 2020*). Although the production and use of PCBs were banned since the 1970s, their presence in the atmosphere and hydrosphere should be monitored, particularly around the E-waste dump sites (*Prithiviraj & Chakraborty, 2020*). Due to their characteristic high persistence and bioaccumulation, PCBs are harmful to both animals and humans, even at low levels (*Elnar et al., 2015*). The toxic effects of DL-PCBs are primarily implemented via activation of the aryl hydrocarbon receptor (AhR) pathway (*Du et al., 2019*; *Fabelova et al., 2019*). In the air, PCBs mainly exist as low-chlorinated congeners (*Halsall et al., 1995*). Notably, NDL low-chlorinated PCBs induce higher oxidative stress levels (*Pessah et al., 2019*) and are more toxic than DL PCBs (*Hansen, 1998*). However, the mechanisms of toxicity induction by NDL PCBs are obscure. The presence of PCB52, known as an indicator PCB, has been demonstrated in food, human blood and milk (*Brunelli et al., 2012*). Exposure to PCB52 causes alterations in motor activity (*Boix et al., 2011*), impairment of neurodevelopment (*Pessah et al., 2019*), and an impaired sense of hearing (*Fabelova et al., 2019*). In vitro exposure to PCB52 induces excessive generation of reactive oxygen species (ROS) in hepatocytes (*Wang et al., 2018*). Subsequently, the high ROS levels increase the release of inflammatory mediators (*Song et al., 2020*), through their effect on toll-like receptors (TLRs)/myeloid differentiation primary response 88 (MyD88) pathway (*Steckling et al., 2020*). These effects can also be executed via the Kelch-like ECH-associated protein 1(Keap1)/ nuclear factor-erythroid 2-related factor (Nrf2)/heme oxygenase-1 (HO-1) pathway, which has both antioxidant and anti-inflammatory effects (*Nguyen, Nioi & Pickett, 2009*; *Song et al., 2020*; *Yamamoto, Kensler & Motohashi, 2018*). Moreover, MyD88-dependent TLR4 mediates accumulation of p62 (also known as SQSTM1), which plays an important role in degradation of autophagosomes (*Fujita et al., 2011*). Aberrant accumulation of p62 results in entrapment of Keap1 and activation of Nrf2 (*Komatsu et al., 2010*).

Overgeneration of ROS is contributed to PCBs-induced hepatotoxicity (*Wang et al., 2018*). ROS accumulation can result in releases of inflammatory mediators, which take an important role in hepatotoxicity (*Khan, Ullah & Nabavi, 2019*). We hypothesized that PCBs-induced excessive generation of ROS and ROS-initiated releases of inflammatory mediators might be responsible for PCBs-induced hepatotoxicity. In the present study, rat (Brl-3A) and human hepatocytes (L-02) were pre-treated with N-acetylcysteine (NAC), a scavenger of ROS, and then exposed to PCB52. The effect of NAC on the rats and human hepatocytes was assessed by determining cell viability, ROS levels, malondialdehyde (MDA) contents, inflammatory, and the activity of the Keap1/Nrf2 pathway. Our findings could provide a novel strategy for the prevention and treatment of PCB52-induced hepatotoxicity.

 

## MATERIALS AND METHODS

### Special chemicals and cell lines

PCB52 (2,2′,5,5′-tetrachlorodiphenyl, purity of 99.4%) was procured from J&K Scientific Company (Beijing, China) and dissolved in dimethylsulfoxide (DMSO) to prepare a 100 mM stock solution. N-acetylcysteine (NAC), and 2′,7′-Dichlorodihydrofluorescein diacetate (DCFH-DA) were purchased from Sigma Chemical Co. (St. Louis, MI, USA). A 600 mmol/L stock solution of NAC was prepared by dissolving solid NAC in saline. Normal human hepatocytes, cell line L-02 (HL-7702) and normal rat hepatocytes, cell line Brl-3A were obtained from the Typical Culture Preservation Commission Cell Bank, Chinese Academy of Sciences, Shanghai, China.

### Cell culture and treatments

Human L-02 cells were cultured in Roswell Park Memorial Institute (RPMI) 1640 medium (Gibco, Dublin, Ireland) supplemented with 15% fetal bovine serum (FBS) and 100 U/ml penicillin-streptomycin. Rat Brl-3A cells were cultured in Dulbecco minimum essential medium (DMEM; Gibco, Dublin, Ireland) supplemented with 3% FBS and 100 U/ml penicillin-streptomycin. Cells were cultured at 37 °C in a thermostatic incubator (Thermo Fisher Scientific Inc., Waltham, MA, USA) infused with carbon dioxide concentration 5%, and 95% saturated atmospheric humidity.

Cells (passage 10) were randomly divided into four groups, including the control, PCB52, NAC, and NAC + PCB52 groups. Following the results of our preliminary experiments (Fig. S1), Brl-3A cells were pretreated with 3 mM NAC for 12 h, while human L-02 cells were pretreated with one mM NAC for 6 h (Gu et al., 2018). The cells were then exposed to 40 µM PCB52 for 48 h and used for subsequent experiments. PCB52 exerted dose- and time-dependent effects (Wang et al., 2018). The concentration of PCB52 and the exposure duration were chosen based on our previous study (Wang et al., 2018) and the changes of cell viabilities in our preliminary experiments.

### Determination of cell viability

Cell viability was determined using the cell counting kit-8 (CCK-8; Dojindo, Kumamoto, Japan) according to the manufacturer's instructions. In brief, cells were first seeded on a 96-well plate. The culture medium was replaced with medium containing PCB52 concentrations of 0, 10, 20, 40 and 80 µmol/L and incubated for 48 h. Then, the PCB media was replaced with a basal medium containing 10 µl of CCK-8 solution and incubated for 3 h at 37 °C. Finally, the absorbance was assessed at 450 nm using a microplate reader (Bio-rad model 680; BIO-RAD, Hercules, CA, USA).

### Monitoring ROS levels

The cells, both L-02 ($5 \times 10^5$ cells/ml) and Brl-3A ($2 \times 10^5$ cells/ml) were seeded in 25 cm$^2$ canted-neck tissue culture flasks (Corning Inc., Corning, NY, USA) and then treated as previously mentioned. Cells were washed twice with PBS, incubated with 10 µmol/L DCFH-DA for 30 min at 37 °C, digested by 0.25% trypsin and washed three times with

 

ice-cold PBS. DCFH-DA fluorescence was analyzed by flow cytometry (BD LSRFortessa™; Becton, Dickinson and Company, Franklin Lakes, NJ, USA).

## Examinations of MDA contents

Briefly, the MDA levels were evaluated using an MDA assay kit (Nanjing Jiancheng Bioengineering Institute, Nanjing, China), according to the operation manual. Optical densities of the supernatants were read at 532 nm using a microplate reader (Bio-rad model 680; BIO-RAD, Hercules, CA, USA) and the protein concentrations of cells were determined using the Pierce™ BCA Protein Assay Kit (Thermo Fisher Scientific Inc., Waltham, MA, USA). The MDA contents are presented as nmol per mg protein in the bar graph.

## Real-time Quantitative PCR

Total RNA was extracted using TRIzol® Reagent (Thermo Fisher Scientific, Waltham, MA, USA), and the RNA quality and quantity were measured using a NanoDrop2000 Spectrophotometer (Thermo Fisher Scientific, Waltham, MA, USA). Complementary DNA (cDNA) was synthesized from the total RNA using the Prime Script RT Reagent Kit (RR036A; Takara Biotechnology Co., LTD., Shiga, Japan). Primers for tumor necrosis factor (TNF) receptor-associated factor 6 (*Traf6*), myeloid differentiation factor88 (*MyD88*), *TNF*, interleukin (*IL*)*1B*, *Keap1*, *Nfe2l2/Nrf2*, heme oxygenase (*Hmox/HO*)-*1*, NAD(P)H quinone oxidoreductase 1 (*NQO1*), *Sqstm1/P62*, and β-*actin* were designed using the Get Prime software (https://gecftools.epfl.ch/getprime). Real-time Quantitative PCR (RT-qPCR) was conducted using TB Green™ Premix Ex Taq™ GC (RR820A; Takara, Shiga, Japan) on a Light Cycler® 96 System (Roche Life Science, Penzberg, Germany). Relative quantification of gene expression was performed based on the comparative CT ($2^{-\Delta\Delta Ct}$) method. The mRNA expression levels of target genes were normalized to that of the β-*actin* internal standard. Data were presented as fold changes over the controls, which were shown as 1. The sequences of primers used for RT-qPCR are shown in Table S1.

## Western blot analysis

Total protein was extracted by lysis using a RIPA buffer (containing 1 mmol/L PMSF), 30 min incubation in ice, and total proteins were collected. Protein concentrations were determined using a Pierce™ BCA Protein Assay Kit (Thermo Fisher Scientific, Waltham, MA, USA). Proteins (20 μg per sample) were separated by SDS-polyacrylamide gel electrophoresis (PAGE) and transferred to polyvinylidene difluoride (PVDF) membranes (BIO-RAD, Hercules, CA, USA). The membranes were then blocked with 5% milk-Tris-buffered-Tween solution for 2 h at room temperature, incubated with primary antibodies at 4 °C overnight, followed by incubation in the appropriate secondary antibodies. Finally, blots were incubated in enhanced chemiluminescent (ECL) detection reagent (BIO-RAD Laboratories, Inc., Hercules, CA, USA) for 3 min. Densitometric analysis was performed using the Tanon Gel Image System (version 4.2). Band intensities were normalized using GAPDH as an internal control. Data on relative integrated

optical density values of bands are displayed in the form of bar charts, with the control presented as 1. This study used primary antibodies against toll-like receptor 4 (TLR4, dilution 1:5,000, sc293072; Santa Cruz Biotechnology, Dallas, TX, USA), Traf6 (dilution 1:3,000, sc8409; Santa Cruz Biotechnology, Dallas, TX, USA), MyD88 (dilution 1:3,000, sc-74532; Santa Cruz Biotechnology, Dallas, TX, USA), Nrf2 (dilution 1:2,000, sc-13032; Santa Cruz Biotechnology, Dallas, TX, USA), keap1 (dilution 1:3,000, sc-15246; Santa Cruz Biotechnology, Dallas, TX, USA), HO-1 (dilution 1:2,000, sc-390991; Santa Cruz Biotechnology, Dallas, TX, USA), NF-κB p65 (dilution 1:2,000, sc-8008; Santa Cruz Biotechnology, Dallas, TX, USA) and GAPDH (dilution 1:5,000, sc-32233; Santa Cruz Biotechnology, Dallas, TX, USA).

### Statistical analysis

Numeric data are presented as mean ± standard error (SE) of at least three independent replicates. Statistical analysis was conducted using the SPSS statistical software (version 16.0; IBM, Armonk, NY, USA). Differences in mean values between the control and PCB52-treated cells were evaluated by the Independent-Samples $T$-test, while differences between the control, PCB52, NAC, and NAC + PCB52 groups were analyzed by One-Way analysis of variance (ANOVA). A $p$-value < 0.05 was considered statistically significant.

## RESULTS

### NAC pretreatment alleviates PCB52-induced hepatotoxicity

Compared to the control group, 48 h exposure of Brl-3A and L-02 cells to PCB52 reduced cell viabilities in a dose-dependent manner (Fig. S1). The most significant reduction was obtained at PCB52 concentrations of 40 and 80 μM. Consistent with our previous study (*Wang et al., 2018*), a PCB52 concentration of 40 μM was found as the most suitable for further studies.

Compared with the control, no significant changes in Brl-3A cell viabilities were observed following 12 h NAC pre-treatment at concentrations 3, 5, and 10 mM. Pretreatment with NAC at concentrations 3 and 5 mM significantly ameliorated PCB52-induced cytotoxicity in Brl-3A cells (Fig. S2). Accordingly, 3 mM NAC was chosen to perform subsequent analyses. Similar to the findings of our preliminary experiment, NAC pretreatment significantly repressed PCB52-induced hepatotoxicity in Brl-3A cells (Fig. 1A). In L-02 cells also, cell viability significantly increased in the NAC + PCB52 group compared with the PCB52 group (Fig. 1B), while no change in cell viability was observed in the NAC group compared with the control (Fig. 1).

### NAC pretreatment mitigated PCB52-induced ROS accumulation and MDA content

The levels of ROS in PCB52-treated cells were significantly higher than in control groups (Figs. 2A and 2B). Compared with the PCB52 group, NAC pretreatment significantly reduced ROS levels in both Brl-3A (Fig. 2A) and L-02 cells (Fig. 2B). However, no significant changes in ROS levels were observed between the NAC group and the control group.

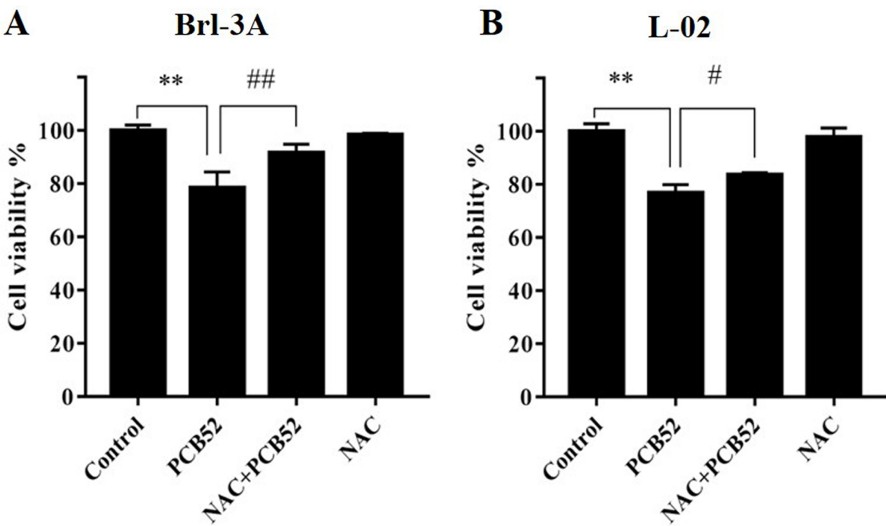

**Figure 1 Detection of cell viabilities by CCK-8 assay.** N-Acetylcysteine (NAC) pretreatment significantly attenuated PCB52-induced cytotoxicity in both Brl-3A (A) and L-02 cells (B), while no significant changes in cell viability were observed in the NAC group compared with the control. $^{**}p < 0.01$, compared with the control; $^{#}p < 0.05$, $^{##}p < 0.01$, compared with the PCB52 group.

Moreover, MDA contents in PCB52-treated Brl-3A and L-02 cells were considerably higher than in control groups (Figs. 2C and 2D). Consistently, NAC pretreatment significantly reduced MDA content compared to the PCB52 group. However, no significant differences in MDA content were observed between the NAC group and the control group.

## NAC pretreatment reduces PCB52-induced inflammatory response

In Brl 3A cells, compared to the control group, the expression of *Traf6*, *MyD88*, and *Tnf* were significantly increased in PCB52 group (Fig. 3A), indicating activation of inflammatory pathways. Also, the expression of *Hmox1*, *Nqo1*, and *Sqstm1* was significantly upregulated by exposure to PCB52, while no significant changes were detected in the expression levels of *Keap1* and *Nfe2l2* (Fig. 3B). NAC pretreatment significantly reduced expression of *Traf6*, *MyD88*, and *Tnf* as compared to the PCB52 group (Fig. 3A), while no significant changes in the expression of *Keap1*, *Nfe2l2*, *Hmox1*, *Nqo1*, and *Sqstm1* were observed (Fig. 3B). Besides, no significant changes in gene expression were observed between the NAC group and the control group. Downregulated expression of *Hmox1* in the NAC + PCB52 group could suggest a reduction of a compensatory response to an oxidative and inflammatory reaction. However, no significant changes in the expression of *Keap1*, *Nfe2l2*, *Nqo1*, and *Sqstm1* were observed.

In L-02 cells, compared with the control, the expression of *MYD88, TNF, and IL1B* were significantly upregulated in PCB52 group (Fig. 3C), while no significant changes were detected in the expression levels of *TRAF6, KEAP1, NFE2L2, HMOX1*, and *NQO1* (Fig. 3D). Expression of *MYD88*, *TNF*, and *IL1B* was significantly decreased in the NAC + PCB52 group compared with the PCB52 group (Fig. 3C). However, no significant changes

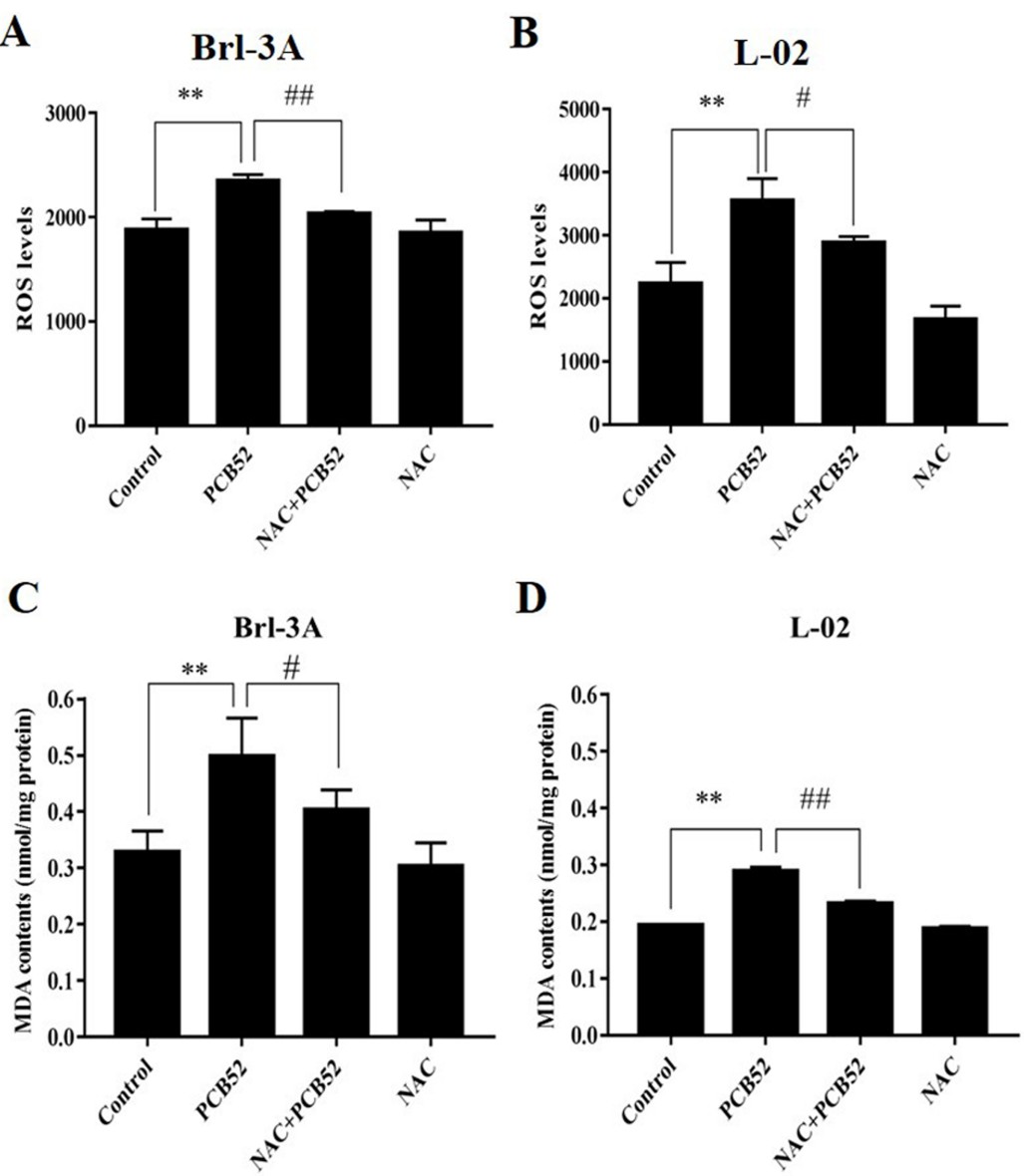

**Figure 2 Examination of intracellular ROS levels and MDA contents.** (A and B) PCB52-induced increase in ROS levels was significantly inhibited by NAC pretreatment in both Brl-3A (A) and L-02 cells (B), while no significant changes in ROS levels were detected in the NAC group compared with the control. (C and D) The increase in MDA contents following the administration of PCB52 was significantly repressed by pretreatment with NAC in both Brl-3A (C) and L-02 cells (D). No significant changes in MDA contents were observed in the NAC group compared with the control. $^{**}p < 0.01$, compared with the control; $^{#}p < 0.05$, $^{##}p < 0.01$, compared to the PCB52 group.

in the expression of *TRAF6*, *KEAP1*, *NFE2L2*, *HMOX1*, and *NQO1* were observed between the two groups. Also, no significant difference was found in the NAC group compared with the control group.

Consistent with the RNA levels, western blot analysis showed that the protein expression of TLR4, MyD88, Traf6, and NF-κB p65 was significantly increased in

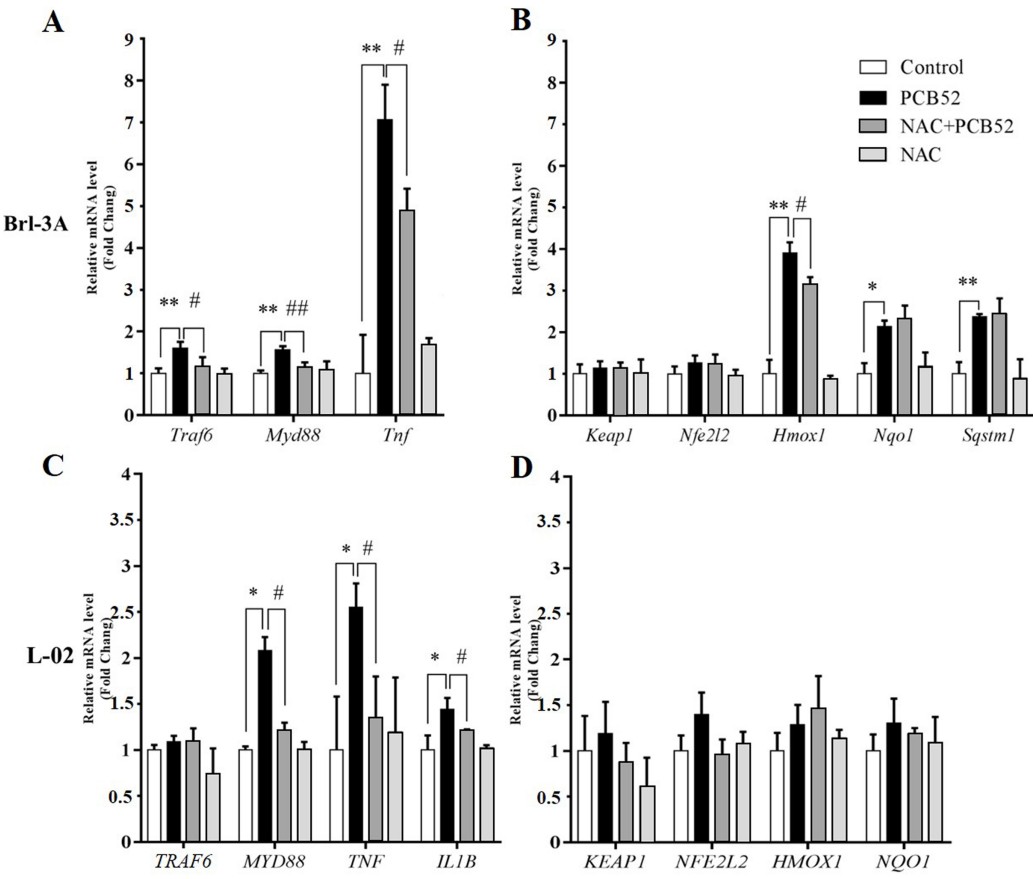

**Figure 3 RT-qPCR analysis for assessment of *Traf6*, *MyD88*, *Tnf*, *IL1B*, *Keap1*, *Nfe2l2*, *Hmox1*, *Nqo1*, and *Sqstm1* gene expression.** (A and B) In Brl-3A cells, the upregulation of *Traf6*, *MyD88*, *Tnf*, and *Hmox1* was significantly reduced in the NAC + PCB52 group relative to the PCB52 group, while no changes in the expression of *Keap1*, *Nfe2l2*, *Nqo1*, and *Sqstm1* were observed. Compared with the control, no changes in gene expression patterns were detected in the NAC group. (C and D) In L-02 cells, expression of *MYD88*, *TNF*, and *IL1B* were significantly dampened in NAC + PCB52 group, relative to the PCB52 group. Compared with the control, no changes in gene expression patterns were detected in the NAC group. $*p < 0.05$, $**p < 0.01$, compared with the control; $\#p < 0.05$, $\#\#p < 0.01$, compared with the PCB52 group.

PCB52-treated Brl-3A cells, as compared with the control (Figs. 4A–4E). However, no significant changes were observed in the protein expression of keap1, Nrf2, and HO-1 (Figs. 4A and 4F–4H). Furthermore, overexpression of TLR4, MyD88, Traf6, and NF-κB p65 proteins was also observed in PCB52-treated L-02 cells (Figs. 4I–4M). Meanwhile, no significant changes in the expression of keap1, Nrf2, and HO-1 proteins were found (Figs. 4I and 4N–4P), which concurs with the observations made in Brl-3A cells. Furthermore, expression of TLR4, MyD88, Traf6, and NF-κB p65 proteins was significantly repressed in the NAC + PCB52 group compared with the PCB52 group in both Brl-3A (Figs. 4A–4E) and L-02 cells (Figs. 4I–4M), suggesting reduced inflammation following NAC pretreatment. However, no significant changes in keap1, Nrf2, and HO-1 proteins were observed.
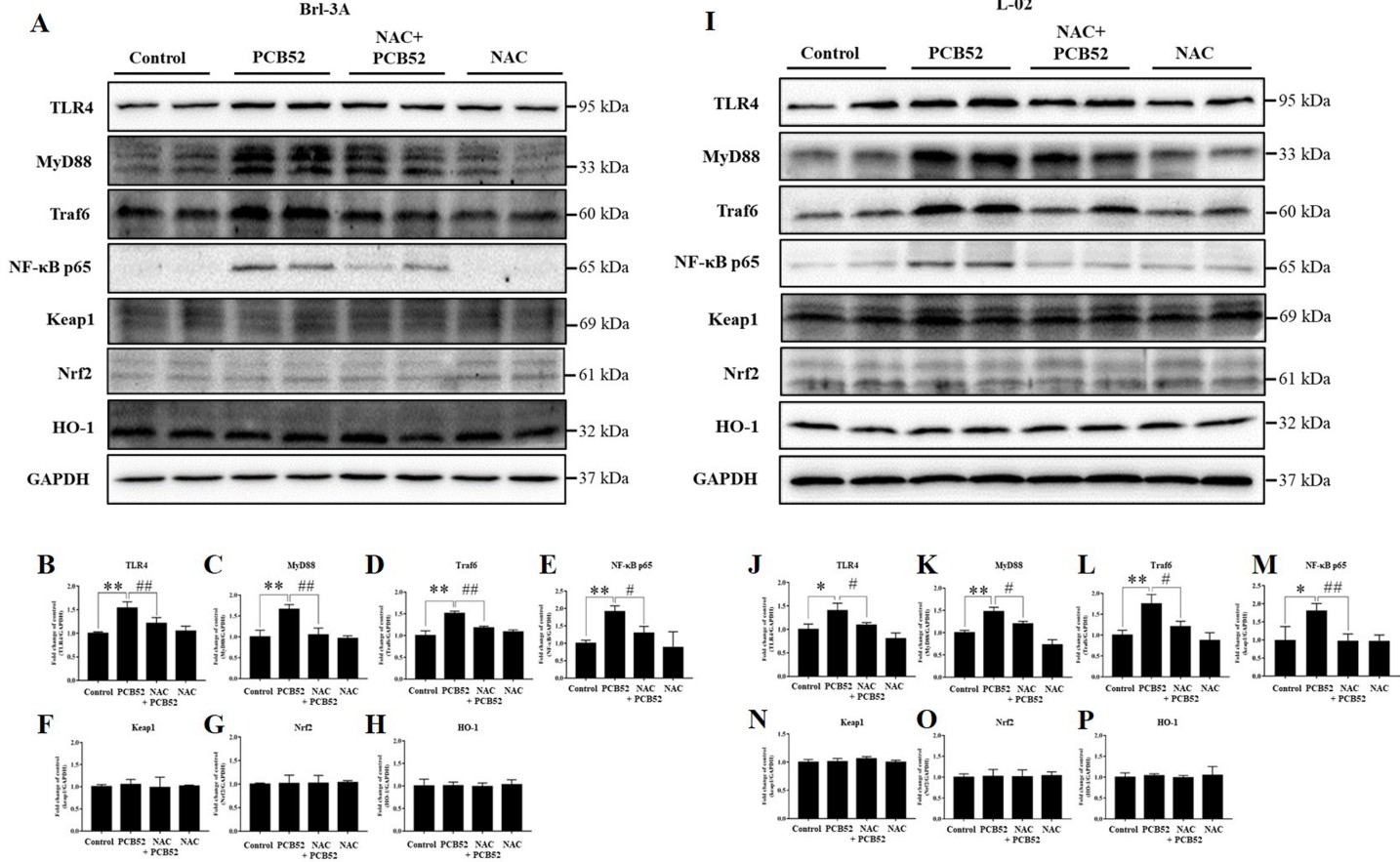

**Figure 4 Western blot analysis for the expression of TLR4, MyD88, Traf6, NF-κB p65, Keap1, Nrf2, and HO-1 proteins.** Increased expression of TLR4, MyD88, Traf6, and NF-κB p65 by PCB52 were significantly repressed by pretreatment with NAC in both Brl-3A (A–E) and L-02 cells (I–M), while no changes in the expression of Keap1, Nrf2, and HO-1 were observed (A, F–H, I, and N–P). Compared with the control, no changes in the expression of these proteins were detected in the NAC group. $^*p < 0.05$, $^{**}p < 0.01$, compared with the control; $^\#p < 0.05$, $^{\#\#}p < 0.01$, compared with the PCB52 group.

## DISCUSSION

Polychlorinated biphenyls (PCBs), particularly low chlorinated PCBs, can currently be detected in our environment and animal tissues (*Wu et al., 2020*). However, the underlying mechanisms of PCB-induced hepatotoxicity, as well as effective treatment for the condition, are indistinct. The findings of the present study showed that PCB52, an indicator of PCBs, induces apparent hepatotoxicity in both Brl-3A and L-02 cells, which is consistent with our previous study results (*Wang et al., 2018*). Moreover, significantly increased ROS levels and MDA contents were detected in cells treated with PCB52. Overgeneration of ROS and dysfunction of the antioxidant system results in an imbalance of redox homeostasis and elevated oxidative stress levels (*Chang et al., 2019*). In this study, elevated ROS levels indicate induction of oxidative stress, which was confirmed by the augmented MDA, a product of polyunsaturated fatty acid (PUFA) peroxidation, which is a component of biological membranes (*Tsikas, 2017*). Furthermore, induction of oxidative stress may result in cell death (*Pirozzi et al., 2020*).

Notably, the overproduction of ROS promotes the release of inflammatory mediators, including TNF-α and IL-1β, via activation of the TLR4/MyD88 pathway (*Gong et al., 2019*; *Shah, Sharma & Banerjee, 2019*; *Steckling et al., 2020*). The transmembrane protein TLR4 is a vital member of the toll-like receptor family, and its interaction with the adaptors MyD88 and TRAF6 are essential for activation of downstream elements and induction of inflammatory response (*Dasu & Jialal, 2013*). In the present study, PCB52 significantly upregulated the expression of *Traf6*, *MyD88*, and *Tnf* in Brl-3A cells, while *MYD88*, *TNF*, and *IL1B* levels were significantly increased in PCB52-treated L-02 cells compared with the control. Consistent with the RNA results, increased expression of TLR4, MyD88, Traf6, and NF-κB p65 proteins were observed in PCB52-treated Brl-3A and L-02 cells, indicating activation of inflammatory responses.

We also found that PCB52 significantly induced the expression of *Hmox1*, *Nqo1*, and *Sqstm1* in Brl-3A cells, while no changes in the gene expression patterns of *Keap1* and *Nfe2l2* were observed. In PCB52-treated L-02 cells, no changes in the expression of *KEAP1*, *NFE2L2*, *HMOX1*, and *NQO1* were observed relative to the control. Moreover, no significant changes in the protein expression of keap1, Nrf2, and HO-1 were detected in PCB52-treated Brl-3A and L-02 cells, suggesting the non-activation of the keap1/Nrf2 pathway, regardless of overproduction of ROS. However, the exact cause of this non-activation needs further investigation. The Keap1/Nrf2 pathway can activate protein HO-1 and is considered to exert anti-oxidant and anti-inflammation effects (*Maamoun et al., 2019*). As an important factor of anti-oxidative stress and anti-inflammatory (*Gozzelino, Jeney & Soares, 2010*; *Willis et al., 1996*), *Hmox-1* gene expression can be mediated by redox-dependent transcription factors (the keap1/Nrf2 system, the transcription repressor Bach1, NF-κB, activating protein-1) and signaling cascades (p38 MAPK, the phosphatidylinositol-3 kinase/AKT pathway, IL-10 and the Jak-STAT pathway, the TLR-4 pathway) (*Naito, Takagi & Higashimura, 2014*; *Paine et al., 2010*). In the present study, increased gene and protein expression of Traf6 and Myd88 as well as overexpression of protein TLR4 might contribute to the up-regulation of gene *Hmox-1* in PCB52-treated Brl-3A cells, although no change of its protein was observed. The result was consistent with previous studies (*Andreas et al., 2018*; *Doberer et al., 2010*). No change of protein expression of HO-1 might be attributed to the mediators of multi signaling cascades and redox-dependent transcription factors as well as the usage and degradation of HO-1 protein (*Andreas et al., 2018*; *Doberer et al., 2010*). In this case, the exact mechanisms of the non-parallel expression of the *Hmox-1* mRNA and its protein need further experiments to clarify. Besides the overproduction of ROS and MDA; therefore, the mute keap1/Nrf2 pathway might exacerbate oxidative stress and cell death.

Nevertheless, pretreatment with NAC, which is a known ROS scavenger, significantly ameliorated PCB52-induced reduction of cell viability, suggesting that elimination of ROS could alleviate PCB52-induced hepatotoxicity. In the NAC + PCB52 group, the ROS level and MDA content were significantly reduced relative to the PCB52 group, indicating repression of oxidative stress. Furthermore, expression of *Traf6*, *MyD88*, and *Tnf* in Brl-3A cells and that of *MYD88*, *TNF* and *IL1B* in L-02 cells as well as TLR4, MyD88, Traf6, and NF-κB p65 proteins in the two cell types were significantly decreased

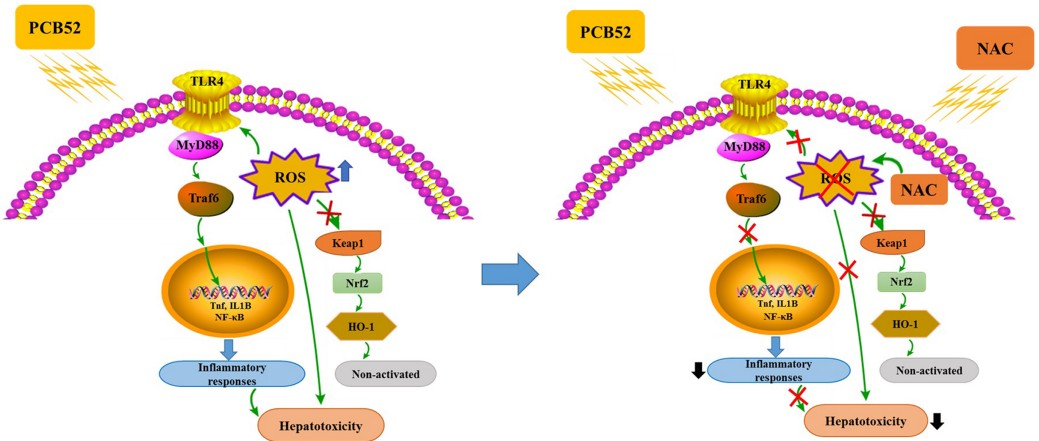

**Figure 5 Schematic diagrams of PCB52-induced hepatotoxicity in vitro and the alleviated effects of NAC.**

following NAC pretreatment. This observation suggests a dampened inflammatory response due to ROS elimination, which concurs with the findings of our previous study (*Li et al., 2020*). In Brl-3A cells also, *Hmox1* was downregulated by pretreatment with NAC, which was consistent with the repressions of the ROS level and MDA content. In addition, compared with the control, gene expression of *Traf6*, *Hmox1*, and *Nqo1* were significant upregulated in PCB52-treated Brl-3A cells but not in L-02 cells, suggesting different sensitivities of cells from different species to PCB52 exposure.

In our previous study (*Wang et al., 2018*), the activation of autophagy resulted in the increase of autophagosomes. The deficit cleavage of autophagosomes induced the blocked autophagic flux, which contributed to PCB52-induced hepatotoxicity. Therefore, the usage of 3-methyladenine (3-MA), an inhibitor of autophagy, alleviated the activated autophagy, the blocked autophagic flux and PCB52-induced hepatotoxicity. Because of the attenuated initiation of autophagy, the positive feedback loop of autophagy and ROS was repressed and ROS levels were significantly decreased by the treatment of 3-MA. However, whether the scavenger-mediated decrease of ROS levels could weaken PCB52-induced hepatotoxicity and the exact mechanisms are unclear. In the present study, the usage of NAC significantly decreased ROS levels and MDA contents, suppressed overexpression of TLR4 pathway, and attenuated PCB52-induced hepatotoxicity, although no change of proteins of keap1/Nrf2 pathway was observed. These results suggested that PCB52 induced hepatotoxicity by initiation of oxidative stress and activation of ROS-mediated inflammatory pathway.

In summary, PCB52-induced oxidative stress, inflammatory responses, and inactive keap1/Nrf2 pathway could be responsible for its hepatotoxicity in vitro (Fig. 5). Notably, all these effects were effectively alleviated by NAC pretreatment, which acts via the elimination of ROS and repression of inflammatory responses (Fig. 5). The findings of this study could provide novel strategies for the treatment and prevention of the hepatotoxic effects associated with exposure to PCBs.

### Funding

The study was supported by the National Natural Science Foundation of China (Grant Nos. 81601641 and 81871526), the Guangdong Natural Science Foundation (Grant Nos. 2014A030310504 and 2020A1515010370), the Scientific Research Foundation for the Returned Overseas Chinese Scholars, the National Education Ministry (Grant No. 2015-311). The funders had no role in study design, data collection and analysis, decision to publish, or preparation of the manuscript.

### Grant Disclosures

The following grant information was disclosed by the authors:
National Natural Science Foundation of China: 81601641 and 81871526.
Guangdong Natural Science Foundation: 2014A030310504 and 2020A1515010370.
National Education Ministry: 2015-311.

### Competing Interests

The authors declare that they have no competing interests.

### Author Contributions

- Wen-Tao Zhou performed the experiments, analyzed the data, prepared figures and/or tables, and approved the final draft.
- Li-Bin Wang performed the experiments, analyzed the data, prepared figures and/or tables, and approved the final draft.
- Hao Yu performed the experiments, prepared figures and/or tables, and approved the final draft.
- Kai-Kai Zhang performed the experiments, prepared figures and/or tables, and approved the final draft.
- Li-Jian Chen performed the experiments, prepared figures and/or tables, and approved the final draft.
- Qi Wang conceived and designed the experiments, authored or reviewed drafts of the paper, and approved the final draft.
- Xiao-Li Xie conceived and designed the experiments, authored or reviewed drafts of the paper, and approved the final draft.

### Data Availability

The raw measurements are available in the Supplemental Files.

### Supplemental Information

Supplemental information for this article can be found online at http://dx.doi.org/10.7717/peerj.9720#supplemental-information.

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
