# Peer review of "N-acetylcysteine alleviates PCB52-induced hepatotoxicity by repressing oxidative stress and inflammatory responses"

_PeerJ, doi:10.7717/peerj.9720_

## Round 0.1 · original submission · Major Revisions

Please see the three Reviewers' comments below. Please address the concerns and suggestions in your rebuttal letter and the manuscript. I look forward to reading your revised manuscript.

Reviewer 1 ·

Basic reporting

The manuscript by Zhou et al. describes the effects of n-acetylcysteine (NAC) on the oxidative stress- and inflammation-induced hepatotoxicity of polychlorinated biphenyl 52 (PCB 52) in two cell lines.

The manuscript and graphics are clear, the structure of the manuscript is appropriate. For more clear understanding, I recommend to make a scheme of genes and respective proteins and their connections (e.g., Keap1/Nrf2 regulating Hmox1 etc) to illustrate the findings of the second part of the manuscript (Figs 3 and 4).

There is a remarkable overlap with the previous publication by the same authors (Wang et al., 2018, https://doi.org/10.1016/j.toxlet.2018.03.002) in terms of overall approach, cell lines and methods. In the previous publication, authors showed the toxicity of PCB 52 to two cell lines, the ability of PCB 52 to induce reactive oxygen species (ROS) and that 3-Methyladenine suppressed ROS. This part is very similar to the first part of the current manuscript. Authors mentioned their previous paper, mostly in the context of PCB 52-induced hepatotoxicity, but they did not explain clearly, what are the knowledge gaps and what is the main hypothesis of the current manuscript compared to the published paper. Also, it would be useful to discuss their current gene expression and Western blot results in the context of the previous findings.

Experimental design

The research hypothesis is unclear.
Methods are mostly well described, only some details are missing (source and passage number of the cell lines; validation of the antibodies).

Validity of the findings

The interpretations that are based on the results of Figs 3 and 4 are inconclusive. For example, in my view, the statement on the "silencing of keap1/Nrf2 pathway" is too speculative. There is no down-regulation of respective mRNAs (Fig 3) and no convincing proof of silencing. In addition, the data on Hmox1 expression on mRNA vs protein level and the explanation of authors is not convincing and should be supported by further experiments.

·

Basic reporting

The article structure is in a good shape.

Experimental design

The experimental design is reasonable.

Validity of the findings

The results are conclusive, but more experiments can be done to better understand the mechanisms.

Additional comments

In this manuscript, the authors investigated the effects of NAC to mitigate the hepatotoxicity mediated by PCBs. Pretreatment with NAC was able to ameliorate the oxidative stress and inflammatory responses in hepatocytes triggered by PCBs. Evidenced by the genes and proteins expression profiles, the authors concluded that the hepatotoxicity was induced mainly through the TLR4-mYD88 pathway, but not Nrf2 pathway. The data presentation is clear, but more discussion is needed to discriminate the results. Some detailed comments for the authors to address.

1. The authors used two hepatocyte lines, one from rat and one from human. The responses in these two cell lines are similar but not the same. What is the rationale to use these two cell lines? The authors should discuss the results.

2. The names of the genes should be consistent throughout the manuscript.

3. In this study, NAC was pretreated with the cells. How about co-exposure of PCB52 with NAC? NAC has a short therapeutic time window evidenced by some in vivo studies. It will be interesting to perform experiments in vitro to understand the best therapeutic time for NAC.

4. The authors found a significant upregulation of TNFα and IL-1β mRNA expression, which indicated the activation of NF-κB. The role of NF-κB activation as down-stream of TLR signaling is implicated in the expression of pro-inflammatory cytokines (TNFα, IL-1β) and assist in the manifestation of inflammation. The authors are encouraged to check the NF-κB activation (phospho-p65) by western blotting.

5. TLR signaling and Nrf2 pathway may crosstalk with each other through p62 driven autophagy for the regulation of inflammatory responses. The authors could discuss more about the communication between these two pathways.

Reviewer 3 ·

Basic reporting

no comment

Experimental design

no comment

Validity of the findings

no comment

Additional comments

In the submitted manuscript, Zhou et al. investigates the mechanism of N-acetylcysteine alleviates PCB52-induced hepatotoxicity. They find that a significant decrease in cell viability and increased ROS levels and malondialdehyde contents in PCB52-treated cells relative to the control, which may be regulated by Keap1/Nrf2 signal pathway and need further study such silencing the Keap1/Nrf2. Overall, the study contributes novel knowledge on the role of NAC in hepatotoxicity caused induced by PCB52. Meanwhile, methods and materials described with sufficient detail and information to replicate. However, there are some points that would need more explanation before publication.


Major comments:
1. Why did the study choose 40 μM of PCB52 for the further experiments? Because it is the IC50 or other reasons.

2. In figure 3, I see some error bar is so high. Is it three independent replicates or three replicates of an experiment? Why did not detect IL-8 in Rat Brl-3A cells and Sqstm1 in human L-02 cells?

3. In figure 4, maybe it’s better to show the protein band size. Because some protein bands are two bands, such as Nrf2 and Keap1.

4. Whether it has time-dependent effect of PCB52 on Brl-3A and L-02 cells, like 24 h, 48 h, 72 h and longer?

5. In the supplementary figure 2, different concentrations of NAC ameliorated PCB52 effects on cell viability in different degree. I wonder why 5 mM and 10 mM of NAC has lower effect than 3 mM under noncytotoxic concentrations?

6. Are there some agonists or antagonists of Keap1/Nrf2 signal pathway for positive or negative control?

7. In the line 132, “And” should be corrected to “and”.

---

## Round 0.2 · accepted · Accept

I look forward to the publication of your article.

Reviewer 1 ·

Basic reporting

Authors answered the questions and significantly improved the manuscript. I propose to accept this manuscript.

Experimental design

Correct.

Validity of the findings

Correct.

·

Basic reporting

No comment

Experimental design

No comment

Validity of the findings

No comment

Additional comments

The questions and comments have been well addressed.

Reviewer 3 ·

Basic reporting

no comment.

Experimental design

no comment.

Validity of the findings

no comment.

Additional comments

In the revised submitted manuscript, the author has a good response to the questions and I don't have other questions. The submission has been greatly improved and is worthy of publication. Finally, I think it is better for author that the change of figures and the revised manuscript was shown in the " Response to the reviwer".